# Towards Photorealistic Video Colorization via Gated Color-Guided Image Diffusion Models

**Jiaxing Li[#]**
Hunan University
Changsha, Hunan, China
lijiaxing0213@gmail.com

**Hongbo Zhao[#]**
Hunan University
Changsha, Hunan, China
hongbozhao@hnu.edu.cn

**Yijun Wang[*]**
Hunan University
Changsha, Hunan, China
wyjun@hnu.edu.cn

**Jianxin Lin[*]**
Hunan University
Changsha, Hunan, China
linjianxin@hnu.edu.cn

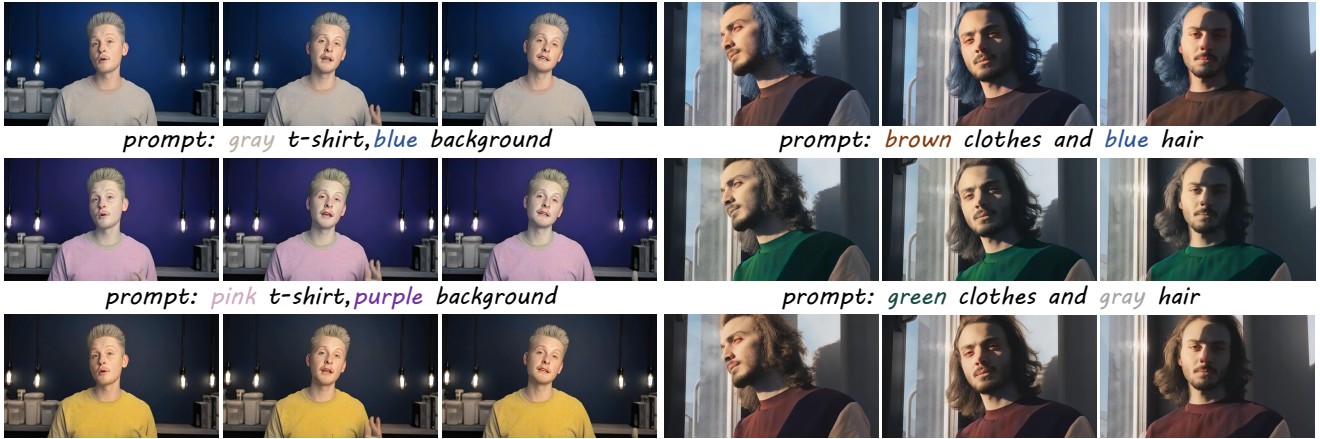

*prompt: gray t-shirt, blue background*

*prompt: brown clothes and blue hair*

*prompt: pink t-shirt, purple background*

*prompt: green clothes and gray hair*

*prompt: yellow t-shirt, black background*

*prompt: red clothes and brown hair*

**Figure 1: Our proposed method can colorize grayscale videos using different prompts while maintaining color consistency between frames.**

## Abstract

Video colorization poses challenging tasks, necessitating structural stability, continuity, and details control in the colors produced. In this paper, based on a pretrained text-to-image model, we introduce the **Gated Color Guidance** module (**GCG**), enabling the model to adaptively perform color propagation or generation according to the structural differences between reference and grayscale frames. Based on this multifunctionality, we propose a novel two-stage coloring strategy. In the first stage, under reference-mask condition, the model autonomously and jointly colors input keyframes in a one-to-many color domain mapping, while temporal coherence constraints are emphasized by modifying the attention mechanism. In the second stage, under reference-guided condition, the model effectively captures the colors of matching structures in the reference, and we further introduce **Sliding Reference Grid** strategy (**SRG**) to merge and extract the color features from multiple frames, providing more stable coloring for the grayscale frames. Through this pipeline, we can achieve high-quality and stable video coloring while maintaining the accuracy of detailed colors. Additionally, the two-stage strategy is flexible and detachable, allowing users to adjust the number of selected reference frames to balance coloring quality and efficiency. Extensive experiments demonstrate that our method significantly outperforms previous state-of-the-art models in both qualitative comparison and quantitative measurement.

## CCS Concepts

• **Computing methodologies** → *Reconstruction*.

## Keywords

Video Colorization, Gated Color Guidance, Diffusion Models

**ACM Reference Format:**
Jiaxing Li[#], Hongbo Zhao[#], Yijun Wang[*], and Jianxin Lin[*]. 2024. Towards Photorealistic Video Colorization via Gated Color-Guided Image Diffusion Models. In *Proceedings of the 32nd ACM International Conference on Multimedia (MM '24), October 28-November 1, 2024, Melbourne, VIC, Australia.* ACM, New York, NY, USA, 10 pages. https://doi.org/10.1145/3664647.3681356

---

[#]Equal contributions.
[*]Corresponding authors.

---

## 1 Introduction

In multimedia processing, video colorization is essential for both aesthetics and cultural heritage restoration. The task involves enriching grayscale videos with true-to-life colors, necessitating precision and temporal continuity, while maintaining vividness of colors and stable detail control.

Previous methods of video colorization can broadly be categorized into three types: The first solution involves coloring each

Jiaxing Li, Hongbo Zhao, Yijun Wang, Jianxin Lin

frame independently and then applying temporal smoothing [5, 30, 32, 38]. This method is heavily reliant on the original coloring results, and the smoothing tends to darken the footage, reducing its color richness. The second solution relies on single frame colorization and color propagation [10, 19, 40, 56] . This approach is prone to the accumulation of errors, causing the hues in later frames to gradually deviate from the original color domain. The third solution encompasses fully-automatic video colorization [29, 31, 48, 70] ; however, the characteristic of this category is a lack of detailed processing. It struggles with samples that include complex movements, often resulting in color artifacts, and generally fails to achieve a rich color palette.

Our work adapts the diffusion model to the task of video colorization, aiming to address the aforementioned issues. One consideration is to adopt text-to-video (T2V) generation methods [14, 15, 46] using extensive text-video datasets for high-quality colorization. However, these methods are cost-prohibitive and unsuitable for widespread content creation. Thus, we prefer fine-tuning on pre-trained text-to-image (T2I) diffusion models. [55, 67] provide valuable templates, expanding self-attention to encompass all frames, allowing a single attention query to access key-value information from all frames and share global information for overall temporal consistency. But this global query operation increases the computational cost during inference, potentially lengthening the colorization time, which could reduce competitiveness in video colorization tasks that have strong structural cues. Based on these analyses, we aim to fully leverage the pairing information of color and structure in the video colorization task. We intend to maintain temporal consistency not solely through global queries but by aggregating information across multiple frames using different strategies. Moreover, previous work reminds us that we should pay more attention to coloring detailed areas and issues such as color leakage.

We propose a new video colorization pipeline, first jointly autocolorizing selected keyframes and then mapping remaining frames to the corresponding color domains of these keyframes, culminating in full video coloring. This requires our model to adaptively perform color propagation or generation based on the differences between the input reference frames and grayscale frames. Therefore, we introduce **Gated Color Guidance** module (**GCG**) into the pre-trained T2I diffusion model, using gating mechanisms to achieve high-quality colorization under different conditions. In detail, under reference-mask condition, it autonomously colorizes input grayscale frames in a one-to-many color domain mapping; under reference-guided condition, it effectively captures and applies the color palette of the reference to grayscale frames in a one-to-one mapping. Moreover, when the reference is available and valid, this gated attention selects structure-color matching information from the reference, filtering out mismatched color information to prevent redundant feature aggregation and stabilize detail colorization.

Specifically, our two-stage coloring strategy involves: in the first stage, key frames are jointly colorized under reference-mask conditions. By employing an extended attention mechanism, these key frames share global information amongst themselves to ensure consistency in colorization; in the second stage, we further explore full video coloring. Differing from the first stage, where each attention operation was expanded to a global query, trading time for

color consistency, we propose a data augmentation method specifically designed for video colorization tasks, termed the **Sliding Reference Grid** (**SRG**). Here, unlike [23], which directly edits the combined grid, we use the grids from multiple video frames as a reference to guide the colorization of images, akin to a sliding window approach. This allows colorizing frames to obtain information from different frames at various time steps, achieving a more stable colorization effect. During fine-tuning, we provide the model with three kinds of inputs: reference-mask, single reference image, and reference grid to foster its capacity for generalization, and through our method, the model is capable of generating stable and high-quality results, as shown in Figure 1. Using common datasets and evaluation metrics, we provide a comprehensive evaluation of the trained model. Extensive experiments verify the superiority of our model in color accuracy, richness, and temporal consistency for video colorization tasks.

Our contributions are summarized as follows:

- We present a novel framework that addresses the challenge of video coloring, decomposing the colorization steps into Keyframes Joint Coloring and Full Video Coloring, which achieve flexible, stable, and high-quality video coloring.
- We propose the **Gated Color Guidance** module (**GCG**), which adaptively performs color propagation or generation by assessing the gap between the reference and input grayscale frames, thus underpinning our two-stage coloring strategy and ensuring accurate colorization and stable detail control.
- We introduce the **Sliding Reference Grid** strategy (**SRG**), utilizing the diffusion model's emphasis on spatial relationships to enable single grayscale frames to capture information from more frames, producing stable outcomes.

## 2 Related Work

### 2.1 Image Colorization

Early image coloring networks were relatively simple and direct in structure. Similar to other CNN tasks, Zhang et al. [65] proposed an end-to-end network for learning the mapping from grayscale images to quantized chromaticity distributions, thus achieving automatic coloring. With the development of Generative Adversarial Networks (GANs) [9], combining the advantages of GANs with learning semantic category distributions has been used to accomplish coloring tasks [27, 50]. However, these methods still have shortcomings in terms of semantic consistency and color richness. To address these challenges, leveraging Transformer's ability for remote context extraction [20, 53], models have been developed to predict a diverse range of color tokens in an autoregressive manner.

Nevertheless, for coloring tasks, users typically desire the ability to color images according to their preferences. Approaches such as [45] and [8] utilized deep adversarial image synthesis architectures to enable users to provide coloring conditions in the form of scribbles. Yun et al. [62] utilized the global receptive field of Transformers to propagate the scribbles provided by users to relevant regions. Coloring methods based on reference images transfer color information from reference images to grayscale images using the color histogram as a prior [61] or through global [51] or local similarity [6]. Additionally, Manjunatha et al. [39], L-CoDe [54]

and L-CoIns [7] applied language models to coloring network tasks to achieve text-controlled coloring.

## 2.2 Video Colorization

Video colorization, distinct from image colorization, requires further consideration of color continuity between frames and consistency of colors with the same semantics. Some post-processing methods [5, 30, 32, 38] are based on pre-trained image colorization models, where each frame is independently edited and then further corrected for temporal discontinuity through post-processing. Despite reducing flickering, these methods often produce frames with insufficient color continuity and faded colors. Another category of methods [10, 19, 40, 56] involves coloring the first frame individually as an example and then sequentially coloring subsequent frames. While these methods offers some degree of color stability, failures in coloring specific frames can lead to a decrease in the propagation effectiveness of colors. Meanwhile, errors may accumulate over time as the frame position shifts, resulting in a gradual deviation of colors from the authentic domain. Fully-automatic methods [29, 31, 48, 70] integrate image colorization and temporal smoothing together. They directly map grayscale frames to their color embeddings via deep neural networks while considering frame continuity. However, these methods struggle to generate color-rich results, and the required time cost is excessively high. Some other methods [4, 18, 57] rely on additional examples provided by the user for coloring, including propagating user scribbles [33, 60, 69] or attaching colors from a reference image to the rest of the frames [59, 63]. However, these methods heavily depend on the quality of the given examples and their match with the video to be colored, and they are prone to washing out colors in some details.

## 2.3 Colorization based on Diffusion Models

Methods based on Diffusion Models (DMs) are rapidly advancing, and their outstanding generative capabilities have been applied in various domains, such as image generation [35, 41, 43] and image editing [3, 12, 24]. Particularly, with the introduction of Latent Diffusion Models (LDM) [44], a method of controlling image and video colorization through conditional injection into the diffusion process has emerged. Specifically, ControlNet [64] enables various conditions to participate in the denoising process of the diffusion model. It integrates more controllable input conditions into the text-to-image synthesis process using a pre-trained latent diffusion model, significantly enhancing the functionality of the diffusion model framework. Building upon this work, the CtrlColor [36] model proposes highly controllable colorization by leveraging combinations of multi-modal conditions. Liu et al. [37] introduced a dedicated coloring assistance module for DM, which has also been extended to the domain of video colorization. Additionally, a significant amount of notable diffusion model work focuses on video editing [11, 26, 55, 67], which can often be applicable to video colorization. Inspiring works such as Tune-A-Video [55] extend the two-dimensional latent diffusion model (LDM) to three dimensions and introduce spatio-temporal attention. ControlVideo [67] extends full cross-frame attention on the original ControlNet for joint editing of all frames, followed by interleaving frames to ensure result stability. However, considering the mutual information between each frame in an n-frame video, we incur computational costs proportional to $O(n^2)$, where some information is redundant.

In contrast, our work first maps keyframes to the same color distribution through joint editing, and then uses these keyframes as references to colorize the remaining grayscale frames through a color distribution control module. By selecting the number of keyframes as a hyperparameter, we allow users to freely balance between precision and faster generation speed.

## 3 Method

### 3.1 Overview

Our work proposes a framework for stably colorizing grayscale video sequences $V = \{I_i\}_{i=0}^{n}$, which produces high-quality videos with rich colors guided by text prompts. To balance coloring quality and computational efficiency, we employ a two-stage strategy. Initially, under the reference-mask condition, we conduct joint coloring on $m$ frames that are abundant in structural and color information. Subsequently, the remaining grayscale frames are mapped to align with the color domain of the key frames. We leverage the pretrained text-to-image (T2I) model and introduce the Gated Color Guidance Module (GCG) to effectively realize high-quality colorization under different conditions using a gating mechanism (Section 3.2). Subsequently, we will demonstrate the two stages of coloring: in Section 3.3, we outline the strategy for selecting keyframes and how to jointly colorize keyframes under reference-mask condition; in Section 3.4, we discuss how to colorize the full video with the keyframes. We introduce the Sliding Reference Grid as a data augmentation method that allows the frame to be colored to merge and extract color information from more frames without altering the network structure, resulting in more stable outcomes. The overall architecture of the proposed method is shown in Figure 2.

### 3.2 Gated Color Guidance Module

We propose the GCG module with the aspiration of achieving the following in supervised video colorization tasks: when reference frames $I^{ref}$ are available, the model can extract effective and structurally matched color information from the reference, mapping the frame to be colored to the corresponding color domain, where the mapping is one-to-one; in the absence of reference or in the parts not involved by reference, the model can freely colorize in a variety of ways, where this mapping characteristic is one-to-many. This multifunctionality underpins our two-stage strategy during inference, and we hope that color propagation with reference will capture more details. Specifically, we designed the following modules, realizing these functions through fine-tuning on datasets.

*3.2.1* ***Color Guidance Attention***. Firstly, we introduce the Color Guidance Attention(CGA) module to achieve the fundamental color propagation functionality, meaning that we aim to guide the frame to be colored $I_i$ to obtain color features that match the colored reference frame $I^{ref}$. We define the representations of $I_i$, $I^{ref}$, and the grayscale reference $I_g^{ref}$ in the latent space as $g_i$, $z^{ref}$, $g^{ref}$. By modifying the traditional cross-attention mechanism [49], we map the feature representations $g_i$, $g^{ref}$, $z^{ref}$ to query $Q$, key $K$, and value $V$ through linear transformations $W_q$, $W_k$, $W_v$, respectively,

 Jiaxing Li, Hongbo Zhao, Yijun Wang, Jianxin Lin

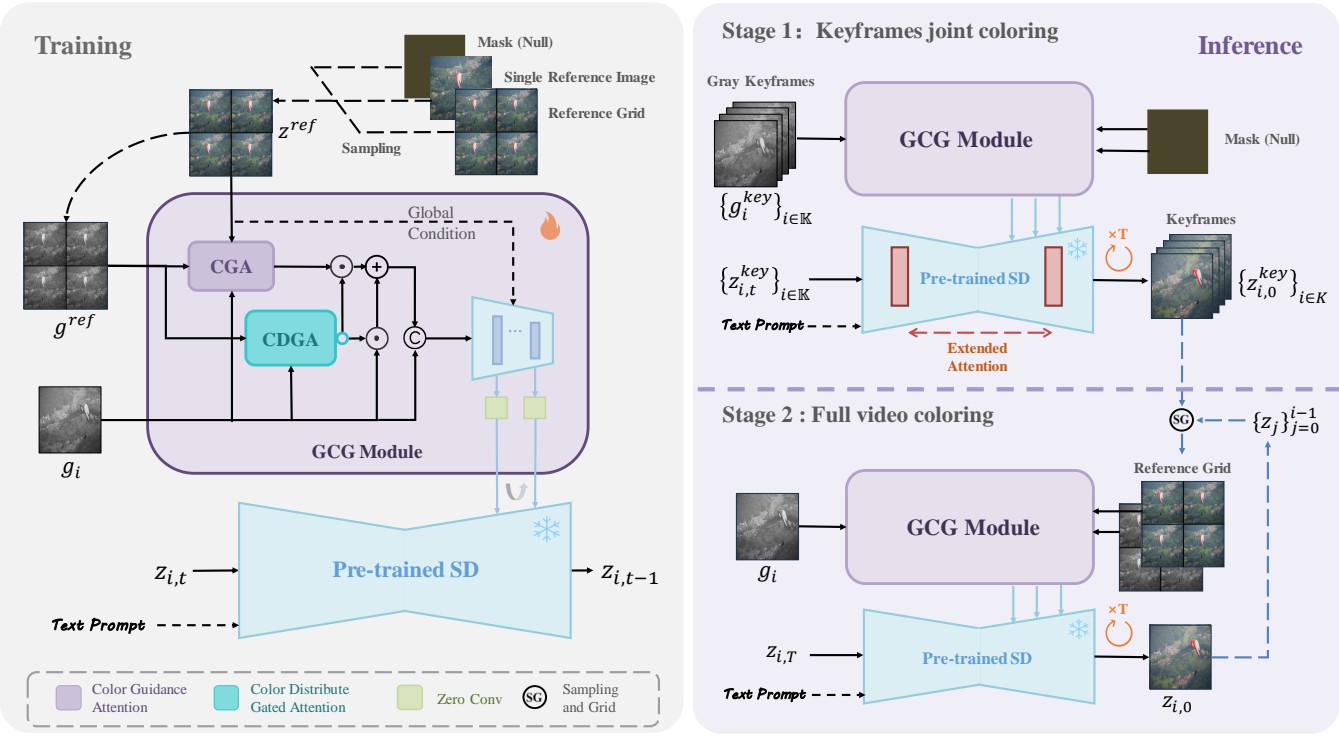

**Figure 2: Framework overview. In the training phase, the Gated Color Guidance module (GCG) serves as an adapter to assist in fine-tuning the pre-trained diffusion model. In the inference phase, we implement a two-stage coloring strategy, i.e., keyframes joint coloring and full video coloring.**

as follows:

$$Q = W_q g_i, K = W_k g^{ref}, V = W_v z^{ref}. \quad (1)$$

Then the result is given by:

$$CGA(g_i, g^{ref}, z^{ref}) = \text{Softmax}\left(\frac{W_q g_i (W_k g^{ref})^T}{\sqrt{d}}\right) W_v z^{ref}, \quad (2)$$

where $\sqrt{d}$ is a scaling factor. Since $g^{ref}$ and $z^{ref}$ maintain strict structural alignment, and $g_i$ and $g^{ref}$ are both grayscale image features, this structure ensures the acquisition of basic color low-dimensional information that matches.

### 3.2.2 *Color Distribution Gated Attention*. Building upon CGA, we further propose the Color Distribution Gated Attention (CDGA) as constraint on the aggregation degree of low-dimensional color information output by CGA with the original information. In CGA, despite the spatial relevance between $g_i$ and $g^{ref}$, they are not perfectly aligned. The aggregation of features from unrelated areas can cause negative impacts. If the difference between $g_i$ and $g^{ref}$ increases, $g_i$ through CGA query obtains more mismatched color information, leading to instability in the final generated effect (which will be discussed in detail in our ablation study in Section 4.5.1). Therefore, an additional control can be applied using a gating mechanism to smooth between $g_i$ and the results queried by $g_i$ through CGA, fading out mismatched color information to prevent redundant feature aggregation.

The implementation of CDGA differs from CGA; we map $g_i$ and $g^{ref}$ onto the Q, K, and V of cross attention as follows:

$$Q = W_q g_i, K = W_k g^{ref}, V = W_v g^{ref}. \quad (3)$$

Then, we map the result range to [0,1], as shown in the following equation:

$$CGDA(g_i, g^{ref}) = \sigma(\text{Softmax}\left(\frac{QK^T}{\sqrt{d}}\right)V)$$
$$= \sigma(\text{Softmax}\left(\frac{W_q g_i (W_k g^{ref})^T}{\sqrt{d}}\right) W_v g^{ref}), \quad (4)$$

where, $\sigma$ denotes the activation function. And unlike the implementation of gated attention in references [1, 35] , here we use CDGA as a regulatory factor.

Through the design of these two modules, our final aggregation result of low-dimensional color reference features is:

$$ColorHint = g_i * (1 - CDGA(g_i, g^{ref})) +$$
$$CGA(g_i, g^{ref}, z^{ref}) * CDGA(g_i, g^{ref}). \quad (5)$$

In this way, by regulating with the factor output by CDGA, we can effectively filter out mismatched structural features, preventing negative impacts from redundant color information aggregation. After obtaining an effective *ColorHint*, we concatenate it with the original grayscale feature $g_i$, serving as input to the UNet copy. In addition, under reference-mask condition, the reference is set as an all-zero mask, i.e., $g^{ref} = z^{ref} = \emptyset$, and the regulation by

CDGA results in *ColorHint* $\approx g_i$, thus coherently achieving the dual functionality mentioned earlier. Furthermore, for the UNet copy, we also utilize the CLIP image encoder to leverage the information of $z^{ref}$ as a global condition [16, 68], implementing cross-attention. Utilizing the shared feature space with the text encoder, it provides semantic features of the reference image, serving as a beneficial initialization to accelerate the entire network training process.

## 3.3 Keyframes Joint Coloring

This section considers how to uniformly colorize keyframes unconditionally. For a grayscale video $V = \{I_i\}_{i=0}^n$ with $n$ frames, we first need to consider how to select the keyframes. Our approach is straightforward: if we need to select a set of $m$ keyframes $\{I_i^{key}\}_{i \in \Bbbk}$, where $\Bbbk \subset U[0, n]$, $|\Bbbk| = m$ and $U$ is the Discrete Uniform Distribution, we just need to satisfy the following equation:

$$\max \sum_{\substack{i,j \in \Bbbk \\ i \neq j}} D(I_i^{key}, I_j^{key}), \tag{6}$$

where $D(\cdot, \cdot)$ is a measure of dissimilarity between the video frames $I_i^{key}$ and $I_j^{key}$. In our actual work, we take the first and the last frames and then select additional $m - 2$ frames to join the keyframe set.

Next, we aim to colorize the obtained set of keyframes $\{I_i^{key}\}_{i \in \Bbbk}$ with reference-mask condition. However, in our proposed GCG module, the gate conditions for reference-mask lean towards one-to-many colorization. Therefore, independently colorizing each keyframe, i.e., $\epsilon'_\theta(z_{i,t}^{key}, t, \tau, \emptyset)$ and $\epsilon'_\theta(z_{j,t}^{key}, t, \tau, \emptyset)$, does not guarantee the consistency and stability of the overall color. To address this issue, following previous work[55, 67], we extend the self-attention module to simultaneously process these keyframes, thereby allowing the set of keyframes to share global color. Specifically, if at time step $t$ during the inference stage, the noisy feature map $z_{i,t}^{key}$ of $I_i$ is mapped to the corresponding query $Q_i$, key $K_i$, and value $V_i$ upon entering the main Unet's self-attention, the original query result for the query $Q_i$ of frame $i$ is:

$$\text{Softmax}\left(\frac{Q_i K_i^T}{\sqrt{d}}\right) V_i. \tag{7}$$

We extend the self-attention module by concatenating the keys and values of all frames, allowing $Q_i$ to query all frames in the keyframe set, resulting in the new attention output:

$$\text{Softmax}\left(\frac{Q_i \mathbf{K}^T}{\sqrt{d}}\right) \mathbf{V}, \tag{8}$$

where $\mathbf{K}$ and $\mathbf{V}$ are respectively the results of concatenating the elements within the sets $\{K_i\}_{i \in \Bbbk}$ and $\{V_i\}_{i \in \Bbbk}$.

In this way, each keyframe's query can aggregate color information from all keyframes, leading to a consistent overall color. It's found that the original attention only queries itself, with an average computation cost equivalent to $o(1)$ for a single frame query, while the computational cost of the extended attention mechanism is $o(m^2)$. Theoretically, we could take all frames of the original video as keyframes, i.e., $m = n$, but this would require longer computation time and more computational resources.

## 3.4 Full Video Coloring

After the keyframes have been colored $\{I_i^{key}\}_{i \in \Bbbk}$, we need to further color the remaining frames $\{I_i^{key}\}_{i \in U[0,n]-\Bbbk}$, while considering how to maintain temporal coherence.

*3.4.1 **Sliding Reference Grid**.* Continuing with the idea of Key joint coloring, we need the noisy feature map of the $i$-th frame at time step $t$ to query information from more frames. We aim to find a method with lower computational cost, instead of expanding the self-attention module, thereby trading time for color consistency. A characteristic of video coloring tasks is the strongly structured reference of grayscale frames, which to some extent allows us to combine images into a grid. Different from [1] where the combined grid replaces all inputs in the pipeline, here the existence of the grid acts as a form of data augmentation, combining only reference frames. Meanwhile, the main Unet backbone processes a single video frame at a time. This prevents the decrease in generation quality due to the combined grid. We call this strategy sliding reference grid. Specifically, we sample from keyframes and already colored video frames, and through a splicing method, they form a new reference image. Then we map its colored and grayscale frames to the inputs $z^{ref}$ and $g^{ref}$ of the GCG, guiding the to-be-colored grayscale image $i$ as follows:

$$\begin{cases} z^{ref} = \text{sampling\&grid}\left(\left\{z_j^{key}\right\}_{j \in \Bbbk}, \{z_k\}_{k=0}^{i-1}\right) \\ g^{ref} = \text{sampling\&grid}\left(\left\{g_j^{key}\right\}_{j \in \Bbbk}, \{g_k\}_{k=0}^{i-1}\right). \end{cases} \tag{9}$$

Regarding the specific sampling strategy, for the latent feature of the i-th frame at time $t$ in the denoising process, i.e., $z_{i,t}$ we aim to satisfy the following optimization condition:

$$\min\left(\frac{\sum_{j=0}^{c_1} D(g_i, g_j^{key})}{\sqrt{t^4 + 1}} + \frac{\sum_{k=0}^{c_2} D(g_i, g_k)}{\sqrt{1 + \frac{1}{t^4}}}\right), \tag{10}$$

$$\text{s.t.} \quad c_1 + c_2 = c,$$

where $D(\cdot, \cdot)$ is defined as in Section 3.3, $c$ denotes the size of the reference grid, meaning that during the inference phase, each frame can reference information from $c$ frames.

Based on this formula, in the early stages of the denoising process, we refer more to key frames to capture global color information, while in the later stages of denoising, we refer more to the neighboring frames of the frame being colorized, to maintain stability and coherence in details between frames.

*3.4.2 **More Details in Practice**.* In order to ensure that our model accurately possesses the capability to adaptively perform color propagation or generation according to the structural differences between reference and grayscale frames, as well as control over detailed colors, we need to fine-tune the GCG module. During the training phase, we will randomly sample from three types of references: reference-mask, single reference image, and reference grid. The loss function for training the model is defined as the Mean Squared Error (MSE) between the predicted noise $\epsilon_\theta$ and the actual noise $\epsilon$:

$$L(\theta) = \mathbb{E}_{t \sim U(1,T), \epsilon \sim \mathcal{N}(0,I)} \|\epsilon - \epsilon_\theta(z_t, t, \tau, g, g^{ref}, z^{ref})\|_2^2, \tag{11}$$

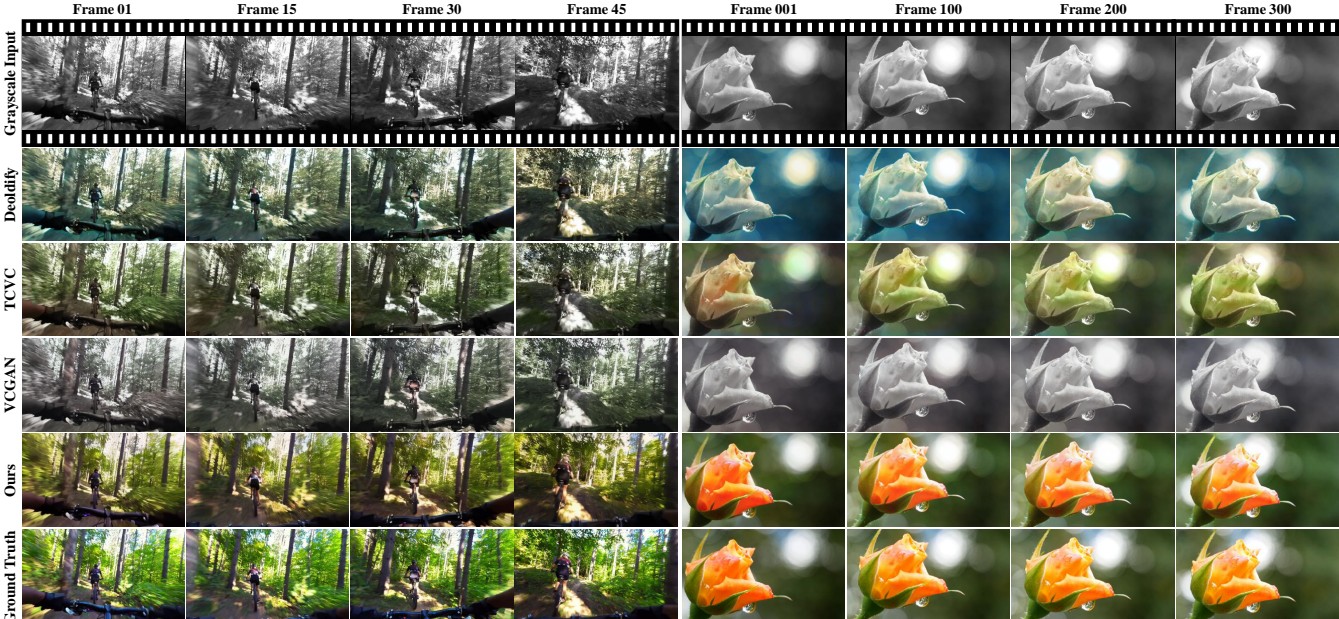

**Figure 3: Qualitative comparison of results from automatic video colorization methods. Our method maintains stable and vibrant coloring in motion and complex scenes *(the left case)* and delivers high-quality, vivid coloring results *(the right case).***

where $U(1, T)$ represents the uniform distribution over the set $\{1, \ldots, T\}$, and $\mathcal{N}(\mu, \Sigma)$ denotes the multivariate Gaussian distribution with mean $\mu$ and covariance $\Sigma$.

Moreover, to ensure that the colorized output maintains structural similarity with the original grayscale input image, we only predicted the *AB* channels from [22]. Employing a strategy akin to [36], we first transform the decoded color image $y$ into LAB space to obtain $y_{LAB}$ and then extract only its AB channel, denoted as $y_{AB}$. Finally, we concatenate $y_{AB}$ with the $L$ channel of the input grayscale image to form the ultimate colorized result $y'$. This approach effectively preserves the original structural features.

## 4 Experiments

### 4.1 Implementation Details

Our model is initialized with the public weights of Stable Diffusion 2.1 [44], following the Adam optimizer [28] with a learning rate set to $5 \times 10^{-5}$. We conducted training for 15 epochs with a batch size of 24. Training was performed at a resolution of $448 \times 256$ using NVIDIA A100 GPU. For the inference process, we accelerated using Denoising Diffusion Implicit Model (DDIM) [47] sampling, set-ting the sampling steps to 32, and setting the guidance scale in the classifier-free guidance approach to 9.0.

During training, we randomly sample from reference-mask, single reference image, and reference grid as reference for colorization, i.e., $z^{ref}$, $g^{ref}$. The reference grid consists of a grid made up of four randomly selected images, including both grayscale and color frames. To ensure diversity in training data, we utilized the HSV model for image color data augmentation, ensuring a variety of colors. For model prompt data, we employed BLIP [34] for text-to-image transformation to obtain prompts. To ensure the model

produces colorized results with a variety of colors, there was a 20% probability that the input is under reference-mask conditions, while colored words were removed from the generated prompts.

### 4.2 Datasets

We employed the Large-scale Diverse Video 3.0 (LDV 3.0) [58] dataset and the DAVIS 2017 [42] dataset as benchmark datasets for training and testing purposes. The LDV 3.0 dataset comprises 365 high-quality videos, encompassing various content categories, motion types, and frame rates. From this, we selected 335 videos for training and reserved the final 30 videos for testing. Additionally, the DAVIS 2017 dataset is a high-quality, high-resolution densely annotated video segmentation dataset, comprising 90 training videos and 30 DAVIS-Test-Dev 2017 testing videos. All video frames were uniformly resized to 448×256 pixels. To obtain grayscale frames, we adopted the approach outlined in [21], utilizing the 'cv2.cvtColor()' operation for conversion to grayscale frames.

### 4.3 Metrics

To evaluate video colorization effectiveness, we consider metrics related to the quality, realism, color vividness, and temporal consistency. PSNR and LPIPS [66] assess image quality and similarity to reality under low-level perception. MUSIQ [25] captures multi-scale structural and statistical regularities, aligning well with human perceptual judgments. CLIP-IQA [52] evaluates semantic distance between images and descriptions, aiding image quality assessment. Colorfulness metric measures color vividness. CDC [38] evaluates color distribution consistency and temporal consistency in videos. FID [13] assesses feature statistics similarity between generated and real images.

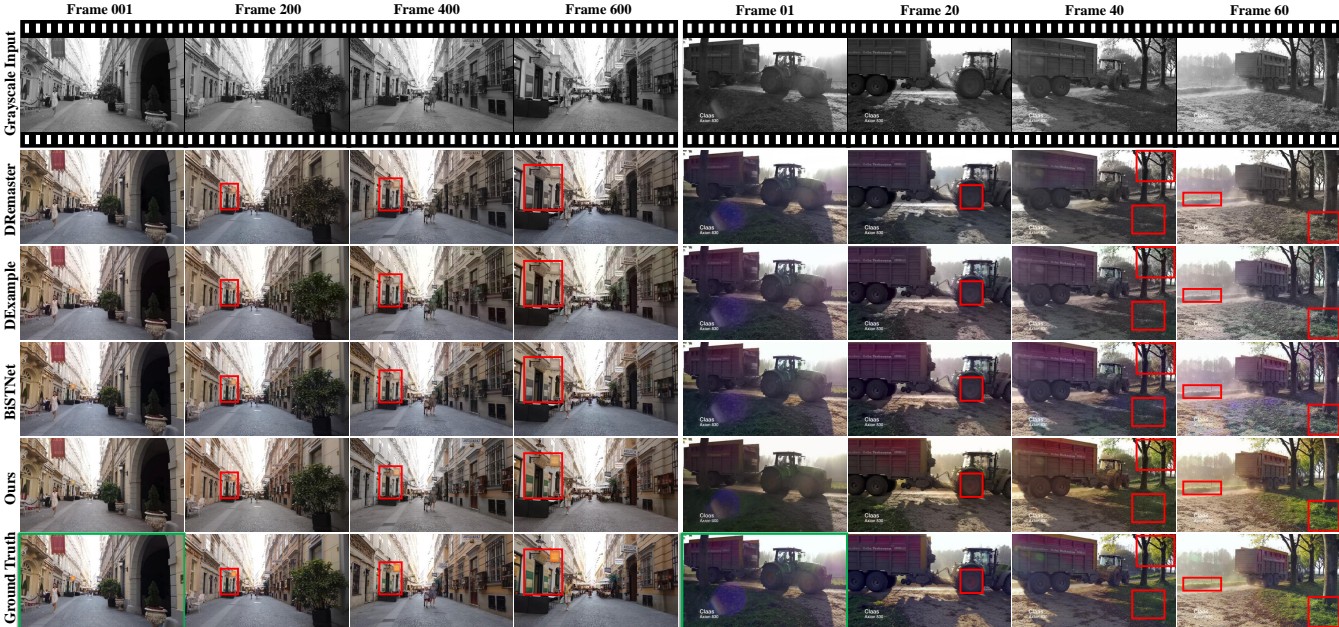

**Figure 4: Qualitative comparison of video colorization results based on reference frames. The exemplar-based video colorization utilizes the first frame of the Ground Truth as the reference frame, indicated by the image with a green border. Here, other methods lack long-term dependencies, causing the video content to gradually darken as the frame position shifts, whereas ours still achieves stable detail control under the same conditions. (Zoom in for details.)**

## 4.4 Comparison

To validate the effectiveness of our method, we will compare it with several state-of-the-art methods, including automatic colorization methods such as Deoldify [2], TCVC [38], VCGAN [70], and exemplar-based methods that require a reference exemplar as guidance, such as DRemaster [17], DExemplar [63], and BiSTNet [59]. For the exemplar-based methods, we will choose the ground truth of the first frame as the reference exemplar. For our model, we also employ the first frame as a reference frame (initially utilizing the Keyframes Joint Coloring method described in Section 3.2, where a reference-mask image is input to generate four consecutive keyframes. Subsequently, these four keyframes are utilized as reference frames, and the strategy outlined in Section 3.3, Full Video Coloring, is applied to complete the full colorization of the grayscale video). Regarding the model prompt input, we similarly utilize BLIP to perform image-to-text transformation for prompt acquisition.

We conducted both quantitative and qualitative evaluations on the synthetic LDV3.0 test datasets and DAVIS-Test-Dev 2017 test datasets, with the results presented in Figure 3, and Figure 4 and Table 1, respectively. In terms of quantitative comparisons, our method outperforms other state-of-the-art methods significantly. Specifically, our method achieves superior results in terms of LPIPS, MUSIQ, FID, and Colorful metrics, indicating that our colorization results are of higher quality and more in line with human perception, demonstrating exceptional performance. It is worth noting that our method is able to utilize the latent color information contained in the pre-trained diffusion models. As shown in Table 1, our method has a clear advantage over all other methods in terms

of the Colorfulness metric. Although our method ranks third in the CDC metric on the synthetic LDV 3.0 test set, this is primarily attributed to our outstanding performance in the Colorful metric, surpassing all other methods. Our results exhibit more vivid and rich colors, which may lead to a higher CDC score. In contrast, the TCVC method, which achieves the optimal CDC score, performs poorly in the Colorful metric, indicating subdued colors that may result in a lower CDC score. Our qualitative evaluation results in Figure 3 also support this observation. In addition, in the qualitative comparison depicted in Figure 3, and Figure 4, our method not only demonstrates advanced capabilities in terms of color coherence and visually pleasing colorization results compared to automatic colorization methods, but also exhibits high color coherence compared to video colorization methods with reference. As the time frames lengthen, scenes may exhibit color characteristics that the reference frames did not possess. In such cases, our method benefits from Color Distribution Gated Attention to achieve consistent automatic colorization. The complete video results are presented in the supplementary material.

## 4.5 Ablation Study

*4.5.1* **Ablation study of the GCG module.** In our model, the GCG module enables the mapping of grayscale frames to color reference frames in a one-to-one or one-to-many manner. In this ablation study, we assess the impact of the GCG module itself, as well as its internal components, Color Guidance Attention (CGA) and Color Distribution Gated Attention (CDGA), on the model's performance. We train the following variants: (1) removing the influence of the GCG module entirely by concatenating grayscale

**Table 1: Quantitative comparisons of our proposed method against state-of-the-art approaches on the synthetic LDV 3.0 test set and the DAVIS-Test-Dev 2017 test set. Here," Ours " represents our final model, " Ours* ", " Ours** ", and " Ours*** " respectively denote the ablation results without CGA and CDGA, with CGA and without CDGA, and without SRG.**

| Dataset | Metrics | Deoldify[2] | TCVC[38] | VCGAN[70] | DRemaster[17] | DExample[63] | BiSTNet[59] | Ours | Ours* | Ours** | Ours*** |
|---|---|---|---|---|---|---|---|---|---|---|---|
| | LPIPS ↓ | 0.1346 | 0.1398 | 0.1565 | 0.1045 | 0.1023 | 0.0806 | **0.0794** | 0.1547 | 0.1136 | 0.0951 |
| | PSNR ↑ | **57.67** | 32.81 | 38.34 | 34.40 | 35.29 | 45.49 | 45.75 | 40.91 | 46.66 | 43.13 |
| | MUSIQ ↑ | 66.88 | 68.72 | 66.23 | 68.52 | 67.50 | 69.40 | **69.57** | 67.45 | 68.35 | 69.72 |
| LDV3 | CLIP-IQA ↑ | 0.5219 | 0.5083 | 0.5360 | 0.5238 | 0.5713 | 0.4807 | **0.5851** | 0.5980 | 0.5908 | 0.5801 |
| | Colorful ↑ | 24.07 | 22.52 | 18.72 | 22.42 | 28.43 | 33.15 | **35.96** | 36.59 | 25.22 | 31.15 |
| | CDC ($\times 10^{-2}$) ↓ | 0.2428 | **0.1724** | 0.3267 | 0.4171 | 0.2236 | 0.1959 | 0.2185 | 0.5871 | 0.3231 | 0.4546 |
| | FID ↓ | 57.20 | 69.91 | 69.91 | 51.48 | 51.09 | 29.47 | **26.84** | 34.50 | 29.62 | 28.23 |
| | LPIPS ↓ | 0.1664 | 0.1768 | 0.1808 | 0.1272 | 0.1016 | 0.0908 | **0.0905** | 0.2185 | 0.1407 | 0.1348 |
| | PSNR ↑ | **57.03** | 32.68 | 38.88 | 33.39 | 44.00 | 43.67 | 47.23 | 42.46 | 47.15 | 46.34 |
| | MUSIQ ↑ | 59.45 | 60.13 | 59.08 | 59.50 | 59.97 | 60.28 | **60.34** | 59.11 | 60.42 | 59.95 |
| DAVIS | CLIP-IQA ↑ | **0.5789** | 0.4918 | 0.5080 | 0.4903 | 0.5415 | 0.5475 | 0.5714 | 0.5399 | 0.5104 | 0.5543 |
| | Colorful ↑ | 20.97 | 22.44 | 15.08 | 21.38 | 26.72 | 26.86 | **29.96** | 29.73 | 23.13 | 27.87 |
| | CDC ($\times 10^{-2}$) ↓ | 0.5776 | **0.3942** | 0.9250 | 0.8492 | 0.5013 | 0.4745 | 0.4624 | 0.8325 | 0.5926 | 0.7383 |
| | FID ↓ | 51.95 | 79.44 | 75.87 | 50.41 | 37.50 | 31.05 | **30.53** | 67.56 | 43.65 | 37.21 |

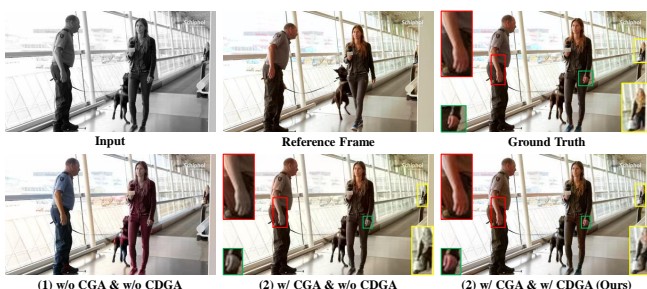

**Figure 5: Ablation study of the GCG module.**

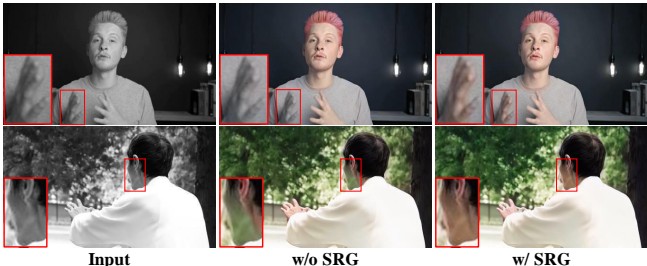

**Figure 6: Ablation study of Sliding Reference Grid.**

frames with reference frames, i.e., w/o CGA & w/o CDGA; (2) using only CGA while excluding the CDGA module's influence, i.e., w/ CGA & w/o CDGA; (3) utilizing the complete GCG module, i.e., w/ CGA & w/ CDGA. We present our experimental results in Figure 5 and Table 1. Observations from (1) show that although the model roughly learns the distribution of color structures and performs well in terms of color structure, the target frames do not effectively learn the corresponding color distribution from the reference frames. Additionally, in (2), although the model can obtain color distribution from the reference frames in most structures, the lack of the CDGA module results in unsatisfactory coloring effects

in the details of the images, as shown in the highlighted box in Figure 5. In contrast, our model achieves adaptive coloring through the GCG module. This not only ensures stable color propagation but also compensates for deficiencies in color distribution within image details by utilizing a gating mechanism.

*4.5.2 **Ablation study of the Sliding Reference Grid (SRG)**.* To validate the effectiveness of SRG, we conducted quantitative and visual comparisons between the results obtained without SRG (i.e., using a single reference image) and those obtained with SRG. This comparison is depicted in Table 1 under the " Ours*** " column and illustrated in Figure 6. When we use a single image as a reference, i.e., w/o SRG, the model may not obtain sufficient color information from the reference, leading to inaccuracies in automatically coloring areas not covered by the reference. For instance, in the upper example of Figure 6, our chosen single reference image does not include the hand, while in the lower example, our chosen single reference image is primarily a frontal face. Therefore, they may exhibit anomalies in the coloring results when new areas appear because they cannot obtain complete color information from the reference. However, by using the Sliding Reference Grid, our model is able to extract as many color structural features as possible from the reference images and previously colored frames, resulting in more natural and continuous outcomes. More details on the ablation experiments are provided in the supplementary materials.

## 5 Conclusion

In this paper, we propose a two-stage coloring pipeline based on the pre-trained T2I model, i.e., Keyframes Joint Coloring and Full Video Coloring, achieving stable and photorealistic video colorization. Our method introduces Gated Color Control module (GCG) to achieve high-quality coloring under various conditions and utilizes Sliding Reference Grid strategy (SRG) for more stable and precise results. Experimental results confirm its superiority over previous state-of-the-art models in terms of color accuracy, richness, and temporal consistency for video colorization tasks.

## Acknowledgments

This research was partially supported by grants from the National Natural Science Foundation of China (Grants No. 62202158, 62206089), the Natural Science Foundation of Hunan Province (Grants No. 2023JJ40167, 2023JJ40178), the science and technology innovation Program of Hunan Province (Grants No. 2023RC3098) and the Fundamental Research Funds for the Central Universities (Grants HNU: 531118010668, 531118010786).

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
