# OpenReview forum: "Towards Photorealistic Video Colorization via Gated Color-Guided Image Diffusion Models"
_acmmm.org/ACMMM/2024/Conference — MM2024 Poster_

### Official Review · Reviewer_apLZ · 2024-05-20

**Rating:** 3
**Confidence:** 2

**Summary:**

This paper introduces a Gated Color Guidance module (GCG), which is used to perform color propagation or generation according to the structural differences between reference frames and grayscale frames.

**Strengths:**

The paper proposes the Gated Color Guidance module (GCG) to generate color proposes a Sliding Reference Grid strategy (SRG), to capture information from more frames. The authors also use common datasets and evaluation metrics to evaluate the trained model.

**Limitations:**

Although the authors propose the GCG model, there are still some shortcomings in the paper.
1. The whole effect of the proposed method is not good enough. For example, in Figure 2, the color of the collar in the second and third rows on the left are unreasonable, and the colors of the eyes, nose and mouth in the second row are distorted.
2. Several kinds of mask are mentioned in the paper, but there are no reasonable explanations and comparisons, and there are no visual results. Therefore, it's not easy for readers to understand the meanings of these masks.

**Suitability:**

2

---

### Official Review · Reviewer_mrSA · 2024-05-22

**Rating:** 4
**Confidence:** 2

**Summary:**

The paper introduces a novel method for video colorization with the Gated Color-Guidance module using Image Diffusion Models. In the training phase, the Gated Color Guidance module serves as an adapter in fine-tuning the pre-trained diffusion model. The inference phase is proposed with two-stage coloring strategy: the first stage is to autonomously colorizes keyframes and the second stage applies color using a reference-guided approach to achieve high-quality colorization. The method leverages diffusion models adapted for video, and the experiments verified the model’s superiority in color accuracy, richness, and temporal consistency for video colorization tasks.

**Strengths:**

- **Novelty**: The approach introduces a new Gated Color Guidance module for video colorization, and decompose the colorization steps into Keyframes Joint Coloring and Full Video Coloring, which achieve accurate colorization and stable control.
- **Theoretical and Technical Correctness**: The paper provides a clear and methodical description of the techniques used, including a Color Guidance Attention module and a Color Distribution Gated Attention module.
- **Adequate Evaluation**: The method is extensively evaluated against existing state-of-the-art models using qualitative and quantitative measures with two datasets, showing superior performance in terms of color accuracy and richness.
- **Clarity**: The paper is well-structured, with clear explanations and supportive figures that illustrate the effectively.
- **Applications**: The technique has practical implications for enhancing grayscale video content, potentially useful in media restoration.

**Limitations:**

- It is not clear of comparison or discussion on time cost to other methods, which is mentioned as a shortcoming in related works.

**Suitability:**

3

---

### Official Review · Reviewer_PwjR · 2024-05-25

**Rating:** 4
**Confidence:** 3

**Summary:**

The paper introduces a novel two-stage video colorization framework that leverages a pre-trained text-to-image (T2I) model. The method aims to produce high-quality, stable, and temporally coherent colorization for grayscale videos.

The authors propose a Gated Color Guidance module (GCG) that adaptively performs color propagation or generation based on the structural differences between reference and grayscale frames. The first stage of the framework involves autonomously coloring keyframes under a reference-mask condition, with an emphasis on temporal coherence through an enhanced attention mechanism. The second stage employs a Sliding Reference Grid strategy (SRG) to extract color features from multiple frames, leading to more stable coloring.

However, I still have some concerns; please see the Limitations section.

**Strengths:**

The article presents several notable strengths in its approach to video colorization:

1. A significant advantage is the GCG module, which adaptively performs color propagation or generation based on the structural differences between reference frames and grayscale frames. This feature enhances the accuracy and stability of the colorization.

2. The SRG strategy is another advantage as it allows for the extraction of color features from multiple frames, contributing to the stability and quality of the colorization, especially in complex scenes.

3. The model emphasizes temporal coherence, which is crucial for video colorization, ensuring that the colors remain consistent across frames.

4. The method is evaluated using common datasets and metrics, which provides a thorough understanding of its performance and capabilities.

**Limitations:**

However, I still have the following concerns:

1. Why use image diffusion models as priors instead of video diffusion models? Considering that 3D diffusion models can generate more consistent results for videos, the explanation in the paper is insufficient.

2. Lack of discussion on the limitations of the method. For example, the performance of long video generation and issues like error accumulation are not addressed. There is also a lack of results for longer videos.

3. Some Implementation details are missing, such as the guidance scale in classifier-free guidance.

**Suitability:**

3

---

### Official Review · Reviewer_4QBZ · 2024-05-26

**Rating:** 4
**Confidence:** 3

**Summary:**

The authors propose a two-stage framework for video colorization, building upon a text-to-image generation model. The proposed GCG module is used for color propagation, and the SRG module is used to enhance local details. Quantitative and qualitative experiments demonstrate that the proposed framework achieves optimal or near-optimal performance.

**Strengths:**

The authors successfully applied a diffusion generative model to the video colorization task, achieving good quantitative results.

**Limitations:**

1. In line 189, the authors mentioned evaluating temporal consistency, but it is unclear which specific metric was used for this evaluation.
2. Lack of efficiency analysis: The authors mention in the abstract that SRG is used to balance coloring quality and efficiency, but there seems to be no experimental data to support this claim.
3. In the quantitative experiments, the authors used prompts generated by BLIP to guide their method. It is unclear whether the baselines in the comparison can also understand textual prompts, which raises concerns about the fairness of the experiments.

**Suitability:**

3

---

### Meta-Review · Area_Chair_5gmG · 2024-06-27

**Recommendation:** Accept (Poster)
**Confidence:** 5

**Metareview:**

Reviewers acknowledged the novelty and good results of the paper. The rebuttal was successful, with three reviewers raising their scores, and the paper received all accept recommendations. The area chairs agree with this recommendation and are pleased to inform you that your paper has been accepted. The authors are advised to incorporate the reviewers' comments into the camera-ready version. Specifically, there is a remaining concern from Reviewer 4QBZ wrt need for an efficiency comparison and a test using the proposed method without additional prompt input.